# New capsaicin analogs as molecular rulers to define the permissive conformation of the mouse TRPV1 ligand-binding pocket

Simon Vu[1], Vikrant Singh[2], Heike Wulff[2], Vladimir Yarov-Yarovoy[1], Jie Zheng[1]*

[1]Department of Physiology and Membrane Biology, University of California Davis, School of Medicine, Davis, United States; [2]Department of Pharmacology, University of California Davis, School of Medicine, Davis, United States

**Abstract** The capsaicin receptor TRPV1 is an outstanding representative of ligand-gated ion channels in ligand selectivity and sensitivity. However, molecular interactions that stabilize the ligand-binding pocket in its permissive conformation, and how many permissive conformations the ligand-binding pocket may adopt, remain unclear. To answer these questions, we designed a pair of novel capsaicin analogs to increase or decrease the ligand size by about 1.5 Å without altering ligand chemistry. Together with capsaicin, these ligands form a set of molecular rulers for investigating ligand-induced conformational changes. Computational modeling and functional tests revealed that structurally these ligands alternate between drastically different binding poses but stabilize the ligand-binding pocket in nearly identical permissive conformations; functionally, they all yielded a stable open state despite varying potencies. Our study suggests the existence of an optimal ligand-binding pocket conformation for capsaicin-mediated TRPV1 activation gating, and reveals multiple ligand-channel interactions that stabilize this permissive conformation.

## Introduction

TRPV1 is a polymodal nociceptor for a wide range of physical and chemical stimuli (*Caterina et al., 1997*; *Tominaga et al., 1998*). Its unique sensitivity to the pungent chili compound capsaicin underlies spicy and hot sensation, and has been used for topical pain treatment through desensitization (*Derry et al., 2017*; *Mason et al., 2004*). Capsaicin-induced TRPV1 activation represents an outstanding case of ligand-receptor interaction in terms of efficacy (near unity open probability at saturating concentrations), potency ($EC_{50}$ at about 100 nM), and ligand discrimination (closely related compounds acting as either an agonist or an antagonist) (*Yang and Zheng, 2017*). Unlike most well-studied ligand-gated ion channels with extracellular or intracellular ligand-binding domains (*Chen and Gouaux, 1997*; *Meyerson et al., 2014*; *Li et al., 2011*; *Zagotta et al., 2003*), TRPV1's ligand-binding pocket resides within the transmembrane domain, formed by the S3 and S4 segments and the S4-S5 linker from the same subunit, as well as the S5 and S6 segments from a neighboring subunit due to a domain-swapping subunit arrangement (*Cao et al., 2013*; *Gao et al., 2016*; *Liao et al., 2013*; *Figure 1A*). TRPV1 functions as an allosteric protein: binding of capsaicin induces a conformational change in the ligand-binding pocket that allosterically promotes the opening of the adjacent activation gate (*Latorre et al., 2007*; *Matta and Ahern, 2007*; *Cao et al., 2014*; *Yang et al., 2010*; *Jara-Oseguera and Islas, 2013*). Molecular interactions between capsaicin and TRPV1 have been studied by cryo-EM (*Cao et al., 2013*; *Gao et al., 2016*; *Liao et al., 2013*) and a combination of computational modeling and functional validations (*Yang et al., 2015*; *Yang et al., 2018*; *Elokely et al., 2016*; *Darré and Domene, 2015*). These recent studies revealed structural mechanisms underlying high-affinity binding and exquisite discrimination between structurally similar ligands. Yet, how capsaicin effectively induces the activation conformational change in the ligand-

*For correspondence:
jzheng@ucdavis.edu

**Competing interests:** The authors declare that no competing interests exist.

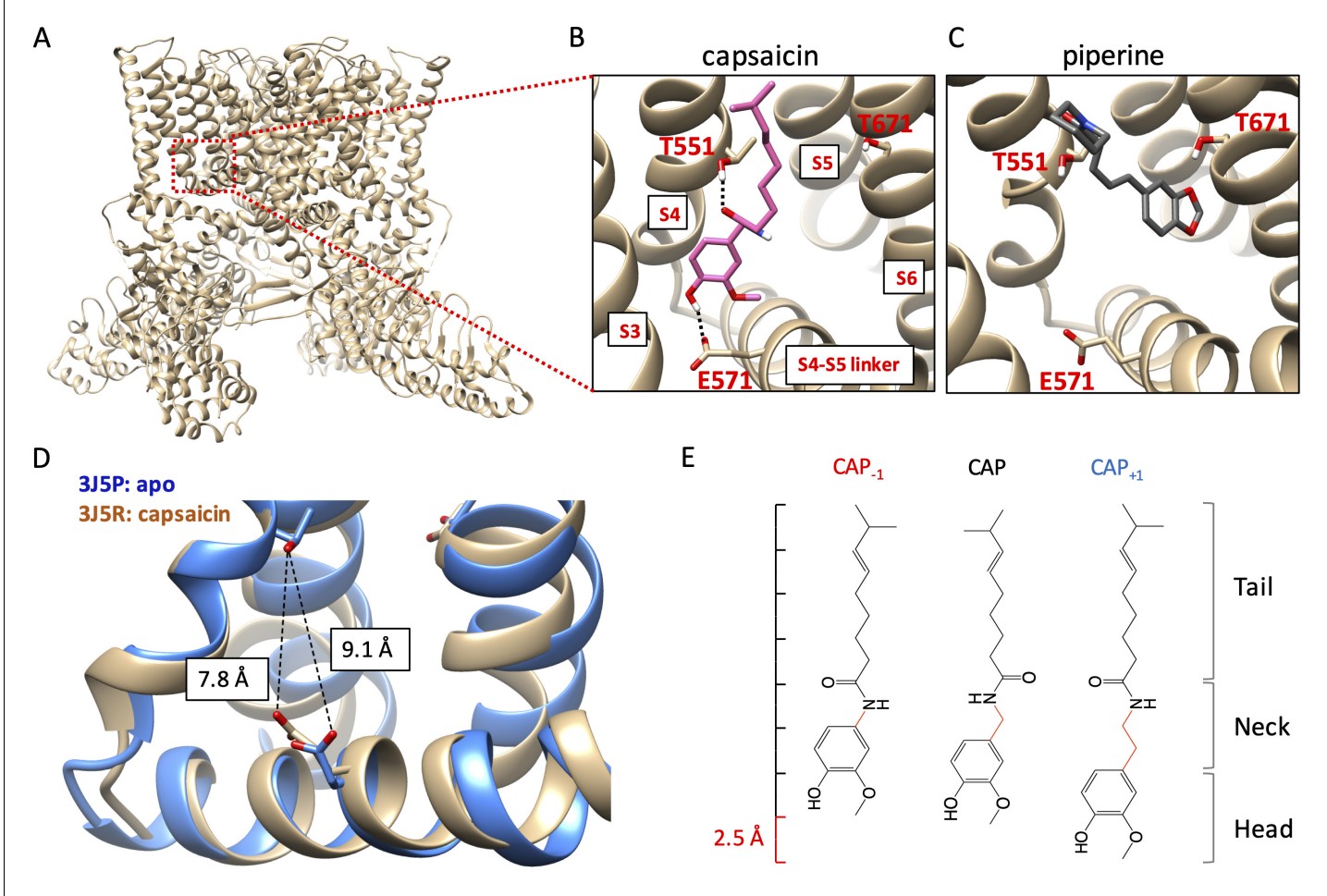

**Figure 1.** Novel capsaicin analogs for the study of TRPV1 ligand-binding pocket. (**A–C**) Characteristics of the ligand-binding pocket with distinct vanilloid binding poses. (**D**) Distance between the two capsaicin hydrogen bonding residues T551 and E571 in the apo state (blue, PDB: 3J5P) and the capsaicin-bound state (brown, PDB: 3J5R). (**E**) Molecular structure of capsaicin and its analogs, with differences in the neck highlighted in red. A rough scale bar is shown to the left.

binding pocket remains less clear. It is also unknown whether diverse capsaicin-derived TRPV1 ligands produces the same or distinct permissive conformations in the ligand-binding pocket.

Activation of TRPV1 can be induced by capsaicin and related pungent plant compounds in two different ligand poses inside the ligand-binding pocket (*Figure 1A*). The highly efficacious mode of activation, induced by capsaicin and its closely related analogs in gingers (e.g. shogaol and gingerol), involves interactions with the S4-S5 linker (*Gao et al., 2016*; *Yang et al., 2015*; *Elokely et al., 2016*; *Darré and Domene, 2015*; *Yin et al., 2019*). In this 'head-down tail-up' vertical pose, the bound ligand bridges the S4 segment and the S4-S5 linker with two hydrogen bonds, one between the amide group in the capsaicin neck and the hydroxyl group of T551 on the TRPV1 S4 segment (using mTRPV1 amino acid number) and another between the hydroxyl group in the capsaicin head and the carboxyl group of E571 on the S4-S5 linker. Simultaneous formation of these two hydrogen bonds lifts the S4-S5 linker upward and outward, a process analogous to voltage activation of Kv channels (*Jiang et al., 2003*). The S4-S5 linker movement reduces the distance between the hydrogen bond-forming sidechains of T551 and E571 by 1.3 Å, as shown by comparing the cryo-EM structures of TRPV1 in the apo and capsaicin-bound states (*Cao et al., 2013*; *Liao et al., 2013*; *Figure 1B*). Movement of the S4-S5 linker away from the pore leads to subsequent S6 movement and pore opening (*Yang et al., 2015*; *Yang et al., 2018*). A less efficacious ligand pose, used by piperine from black peppers (and zingerone from gingers), involves direct interaction with the S6 segment instead of the S4-S5 linker (*Yin et al., 2019*; *Dong et al., 2019*). In this horizontal pose,

piperine is found to bridge T551 on the S4 segment and T671 in the middle of the S6 segment, which is expected to cause S6 movement and pore opening. For capsaicin activation, movement of S6 also occurs; it couples the S4-S5 linker movement to the opening of the activation gates (*Yang et al., 2018*). In support of the existence of these two distinct ligand poses, preventing hydrogen bond formation with the S4-S5 linker by an E571A mutation strongly affected activation by capsaicin but not piperine, whereas mutations at T671 (to S or V) on S6 disrupted activation by piperine but had little effects on activation by capsaicin (*Yang et al., 2015*; *Dong et al., 2019*). Although direct interaction with S6 by piperine or zingerone does not yield high-efficacy efficacious channel activation, this form of ligand activation is attractive for developing TRPV1 positive allosteric modulators for clinical purposes (*Kaszas et al., 2012*; *Lebovitz et al., 2012*).

In order to understand why different agonists take distinct binding poses, and how these poses allow ligands to stabilize the open conformation of the ligand-binding pocket, we designed two capsaicin analogs with an altered neck length (*Figure 1C*): $Cap_{+1}$ contains one extra -$CH_2$ group between the hydrogen-bond-forming groups, whereas $Cap_{-1}$ contains one less -$CH_2$ group between the hydrogen-bond-forming groups. Changes in the neck length of these compounds (±1 C-C bond length) are compatible to the 1.3 Å movement of the S4-S5 linker from the closed state to the open state. Importantly, these structural changes preserved all the key functional groups involved in direct ligand-channel interactions. Together with capsaicin, these compounds served as a set of molecular rulers for the conformational changes in the ligand-binding pocket. Binding of these capsaicin analogs to TRPV1 as well as their abilities to induce channel activation were investigated by a combination of computational modeling, pharmacological analyses, and mutational tests.

## Results

### Capsaicin analogs strongly activate TRPV1

We recorded current responses of mouse TRPV1 expressed in HEK cells to capsaicin and its two analogs at both macroscopic and single-channel levels (*Figure 2A and C*). Both $Cap_{+1}$ and $Cap_{-1}$ strongly activated TRPV1 by binding to the same binding site as capsaicin (confirmed by competition with capsazepine; see Figure 4B below). As previously reported (*Yang et al., 2015*), capsaicin activated mouse TRPV1 with an $EC_{50}$ value of 0.14 ± 0.01 μM and an open probability (*Po*) of 0.94 (*Figure 2B and F*), comparable to observations from the rat TRPV1 (*Rosenbaum et al., 2004*; *Geron et al., 2018*; *Morales-Lázaro et al., 2016*). $Cap_{-1}$ activated the channel at a similar concentration range compared to capsaicin; however, the maximal currents induced by $Cap_{-1}$ were smaller by about 10% (*Figure 2A*, middle panel). We found that the reduced maximal current amplitude was likely caused by a slightly lower single-channel conductance (*Figure 2C,D,F*). After correction for the conductance difference, macroscopic responses to $CAP_{-1}$ exhibited similar maximal *Po* and $EC_{50}$ values to capsaicin responses (*Figure 2B and F*). High *Po* could be directly observed from $CAP_{-1}$-induced single-channel currents, confirming that $CAP_{-1}$ is highly efficacious (*Figure 2C,E,F*). $Cap_{+1}$ activated TRPV1 to similar single-channel conductance and maximal *Po* as capsaicin (*Figure 2A and C*); however, it took much higher concentrations of $Cap_{+1}$ to reach the maximal *Po* level (*Figure 2B and F*). Therefore, lengthening and shortening the capsaicin neck by one C-C bond length did not disrupt ligand-channel interactions but appeared to have altered the nature of the interactions. Furthermore, it is interesting to observe that lengthening the ligand ($CAP_{+1}$) did not affect the maximal *Po* (as one might expect if the extended ligand is a better fit to the closed-state ligand-binding pocket), but instead reduced potency.

### $Cap_{-1}$ and $Cap_{+1}$ stabilize ligand-binding pocket with alternative binding poses

To understand how changing the neck length could produce the observed effects on agonist binding and subsequent channel activation, we analyzed the binding poses of $Cap_{-1}$ and $Cap_{+1}$ inside the ligand-binding pocket. Using the RosettaLigand application within the Rosetta molecular modeling software suite (*Davis and Baker, 2009*; *Davis et al., 2009*; *Meiler and Baker, 2006*; *Bender et al., 2016*) and the capsaicin-bound cryo-EM structure (PDB index: 3J5R *Cao et al., 2013*) as the starting template, we identified the most stable (lowest energy) binding poses. The capsaicin-bound cryo-EM structure was chosen over the resiniferatoxin (RTX)/double-knot toxin (DkTx)-bound

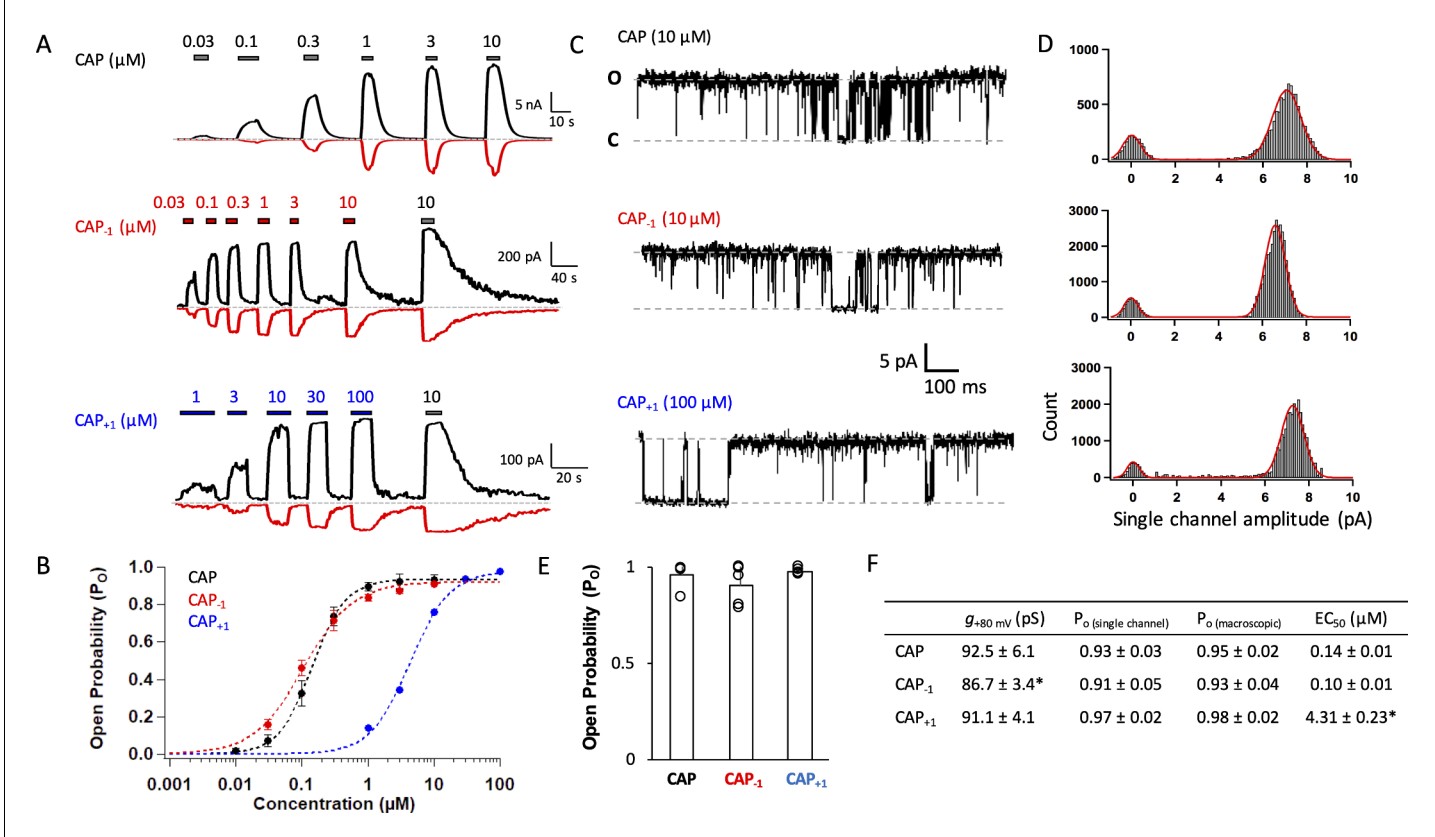

**Figure 2.** Capsaicin analogs with a shorter or longer neck can fully activate TRPV1. (**A**) Representative current traces induced by capsaicin or its analogs recorded at +80 mV (black) and −80 mV (red). (**B**) Concentration-response curves ($n$ = 5–8 cells). (**C**) Representative single-channel traces in the presence of a saturating concentration of capsaicin or its analogs at +80 mV. Upward deflection represents the channel in the open state. (**D**) All-point histogram of single-channel events induced by respective agonists. $n$ = 3–4 cells. (**E**) Comparison of the open probability. (**F**) Conductance, $P_O$ determined from single-channel and macroscopic recordings, and $EC_{50}$ values at +80 mV in the presence of a saturating concentration of each agonist (Cap, 10 μM; Cap$_{-1}$, 10 μM; Cap$_{+1}$, 100 μM; $n$ = 3–5 cells). *, $p<0.05$.

structure (PDB index 5IRX *Gao et al., 2016*) as all agonists in this study were capsaicin derivatives, which are substantially different in molecular structure from RTX. Hence cautions should be applied when extending conclusions of this study to other TRPV1 ligands. Note that annular lipids and PIP2 known to exert a strong influence on TRPV1 activation were not modeled. Similarly, potential involvement of water molecules in mediating ligand-channel interactions was not modeled. These omissions could substantially affect the energetics during the modeling process. Hence, we individually verified all key interactions predicted from structural modeling using functional tests.

Representatives among the top 30 models are illustrated in *Figure 3A and D* for Cap$_{-1}$ and Cap$_{+1}$, respectively. Distributions of average hydrogen bond and Van der Waals interaction energies (in Rosetta energy units) of the top 30 models are also shown (*Figure 3B and E*). Among the top models, we found similar binding poses as that of capsaicin, with Cap$_{-1}$ or Cap$_{+1}$ in a 'head-down tail-up' vertical pose (*Figure 3A and D*, upper panels; see also *Figure 3—figure supplement 1*, left panel). In this orientation, the agonist formed hydrogen bonds with T551 and E571 to stabilize the upward conformation of the S4-S5 linker. This capsaicin-like binding pose could be functionally confirmed. When T551 or E571 was mutated to a nonpolar residue to prevent hydrogen bond formation at these two positions, the concentration-response curves for Cap$_{-1}$ or Cap$_{+1}$ were dramatically affected: we observed a one to two orders of magnitude increase in $EC_{50}$ and a large decrease in *Po_max* (*Figure 3C and F*; *Table 1*).

Interestingly, among the top models we also observed ligand poses distinct from that of capsaicin. For Cap$_{-1}$, Rosetta modeling predicted a semi-horizontal pose in which it forms a hydrogen bond with T671 on S6 using the carbonyl oxygen atom in its neck (*Figure 3A*, lower panel). The

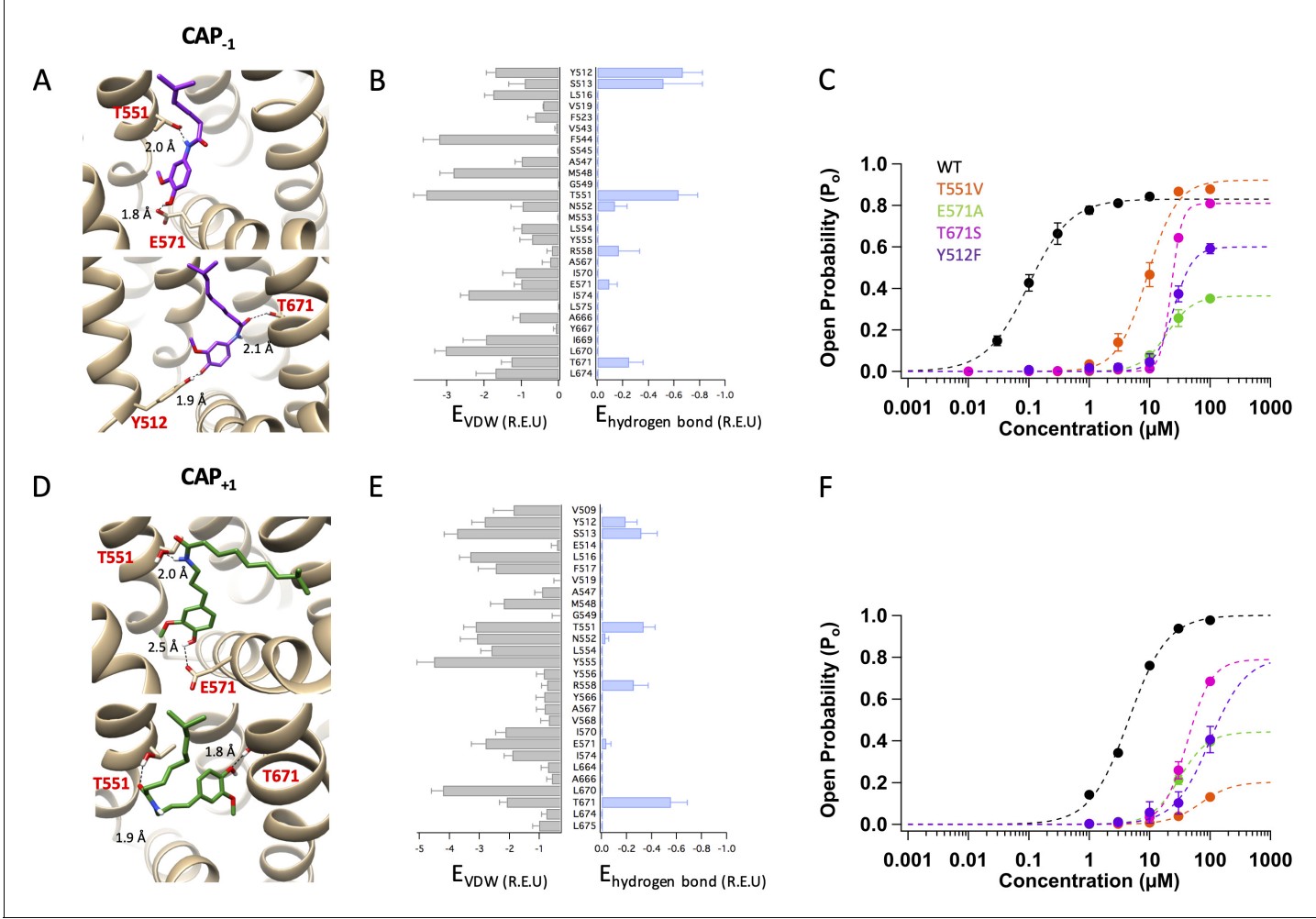

**Figure 3.** Docking of Cap$_{-1}$ (A–C) and Cap$_{+1}$ (D–F) reveals interacting channel residues and binding poses of agonists. (A) Representative binding poses of CAP$_{-1}$ inside the ligand-binding pocket. (B) Energy of predicted VDW and hydrogen bond for each ligand-binding pocket residues. (C) Concentration-response curves of WT and mutant channels in the presence of CAP$_{-1}$. (D) Representative binding poses of CAP$_{+1}$. (E) Energy of predicted VDW and hydrogen bond for each ligand-binding pocket residues. (F) Concentration-response curves of WT and mutant channels in the presence of CAP$_{+1}$. Units of energy are R.E.U. (Rosetta Energy Unit). $n$ = 3–5 cells.

The online version of this article includes the following figure supplement(s) for figure 3:

**Figure supplement 1.** Representatives of the vertical binding poses for Cap$_{-1}$ (left) and the horizontal binding poses for Cap$_{+1}$ (right) among the 30 top models.

**Figure supplement 2.** Representative whole-cell patch-clamp recordings of TRPV1 T671S mutant activated by capsaicin (top left), Cap$_{-1}$ (top right), and Cap$_{+1}$ (bottom left), and the concentration-response curves (bottom right).

**Figure supplement 3.** Representative whole-cell patch-clamp recordings of TRPV1 Y512F mutant activated by capsaicin (top left) and Cap$_{-1}$ (top right), and the concentration-response curves (bottom).

Cap$_{-1}$ head is located in the vicinity of the S2-S3 linker where it interacts with Y512. In this pose, Cap$_{-1}$ directly interacts with S6 just as piperine does (*Dong et al., 2019*). There is, however, an important difference: piperine uses T551 as an anchor point for binding (*Figure 1A*, right panel), whereas Cap$_{-1}$ uses Y512 as the anchor point (*Figure 3A*, lower panel). Functional data were supportive of the presence of this binding pose: conserved mutations T671S and Y512F right-shifted the concentration-response curve by two orders of magnitude (*Figure 3C*; *Figure 3—figure supplements 2* and *3*). In comparison, T671S had no effect on capsaicin activation, whereas Y512F had a minor effect (*Yang et al., 2015*; *Figure 3—figure supplements 2* and *3*). Therefore, it appears that

**Table 1.** Comparison of ligand activation of wildtype and mutant mTRPV1 channels by capaicin analogs.

| | Cap | | | Cap$_{-1}$ | | | Cap$_{+1}$ | | |
|---|---|---|---|---|---|---|---|---|---|
| | EC$_{50}$ (µM) | K | N | EC$_{50}$ (µM) | K | N | EC$_{50}$ (µM) | K | N |
| WT | 0.14 ± 0.01 | 1.82 ± 0.22 | 4 | 0.10 ± 0.01 | 1.20 ± 0.05 | 3 | 4.31 ± 0.23[*] | 1.44 ± 0.11 | 4 |
| T551V | 1.56 ± 0.20[†] | 1.74 ± 0.13 | 5 | 9.13 ± 1.10[*‡] | 1.83 ± 0.31 | 5 | 70.12 ± 7.74[*§] | 1.71 ± 0.10 | 8 |
| E571A | 1.53 ± 0.10[†] | 1.86 ± 0.07 | 4 | 19.31 ± 0.22[*‡] | 2.01 ± 0.02 | 6 | 31.43 ± 0.02[*§] | 1.84 ± 0.11 | 6 |
| I574A | 0.22 ± 0.02[†] | 1.41 ± 0.12 | 3 | 2.80 ± 0.11[*‡] | 1.35 ± 0.23 | 7 | 10.73 ± 0.33[*§] | 1.98 ± 0.12 | 5 |
| T671S | 0.05 ± 0.06 | 0.87 ± 0.11 | 3 | 22.80 ± 0.40[*‡] | 4.94 ± 0.30 | 3 | 35.29 ± 1.19[*§] | 3.16 ± 0.45 | 3 |
| Y512F | 1.12 ± 0.17[†] | 1.35 ± 0.08 | 4 | 25.18 ± 1.18[*‡] | 2.68 ± 0.34 | 3 | 99.48 ± 6.82[*§] | 1.46 ± 0.17 | 3 |

[*] Compared to Capsaicin (Cap).

[†]Compared to WT Cap.

[‡]Compared to WT Cap$_{-1}$.

[§]Compared to WT Cap$_{+1}$.

EC$_{50}$, half maximal effective concentration; K, Hill coefficient; N, number of patches.

Cap$_{-1}$ may take either a vertical binding pose like capsaicin to interact with the S4-S5 linker or a semi-horizontal binding pose similar to piperine to interact with S6.

Cap$_{+1}$ was predicted to also interact with T671 on S6 with a hydrogen bond, which is formed with the oxygen atom in the head hydroxyl group (***Figure 3D***, lower panel). Indeed, binding energy distribution indicates that Cap$_{+1}$ interacts predominantly with T671 instead of E571 on the S4-S5 linker (***Figure 3E***; see also ***Figure 3—figure supplement 1***, right panel). In this horizontal pose, the Cap$_{+1}$ neck forms a hydrogen bond with T551 and establishes VDW interactions with the nearby Y555. In support of the presence of this binding pose, both T671S and T551V mutations right-shifted the concentration-response curve by two orders of magnitude and reduced the maximal *Po* (***Figure 3F***; see also ***Figure 3—figure supplement 2***). We noticed that in this pose the tail of Cap$_{+1}$ is predicted to be unable to fully extend into the hydrophobic region in the upper ligand-binding pocket. For capsaicin, we have previously found that its tail contributes substantially to binding via VDW interactions with the heavily hydrophobic upper part of ligand-binding pocket (***Yang et al., 2015***). Reducing these VDW interactions might be part of the reason that Cap$_{+1}$ binds much less tightly than capsaicin and Cap$_{-1}$. Interestingly, the ginger compound zingerone, which shares the same head and neck structures with shogaol and gingerol but is devoid of a hydrophobic tail, was also found to take the horizontal pose to interact with T551 and T671 even though shogaol and gingerol predominantly take a vertical pose similar to capsaicin (***Yin et al., 2019***).

## Cap$_{-1}$ and Cap$_{+1}$ exhibited altered gating properties

The high *Po* produced by capsaicin and its two analogs, indicating a stable open state of the channel, permitted not only reliable structural modeling but also detailed functional investigation of the ligand-channel interaction mechanism. To better understand the molecular interactions between capsaicin analogs and TRPV1 seen in structural modeling, we first examined their binding properties with two functional tests. We compared their unbinding kinetics to that of capsaicin by measuring the OFF rate from the tail current (***Figure 4—figure supplement 1***). We found that the OFF rates for Cap$_{-1}$ and Cap$_{+1}$ were two to six times faster than that of capsaicin (***Figure 4A***), indicating that binding of these analogs is less stable. Following this observation, we compared stability of bound ligands at the equilibrium state, using competition between the TRPV1 inhibitor capsazepine and capsaicin or its analogs (***Figure 4—figure supplement 2***). Capsazepine has been shown by cryo-EM to occupy the same site as capsaicin (***Gao et al., 2016***). We found that capsazepine could compete off both Cap$_{-1}$ and Cap$_{+1}$ more easily than capsaicin (***Figure 4B***). In summary, both kinetics and equilibrium properties suggested weaker binding of Cap$_{-1}$ and Cap$_{+1}$.

We next examined how well Cap$_{-1}$ and Cap$_{+1}$ induced channel opening when bound to TRPV1. Capsaicin activates TRPV1 through an allosteric mechanism that can be simplified to a three-state model as shown in Figure 4C, top panel (***Zagotta, 2015***). There are two equilibrium constants (free parameters) in the model: $L$ and $K_D$. $L$ represents the equilibrium constant for the activation gating step and can be estimated from the maximal open probability, *Po_max*. Therefore, $L$ can be

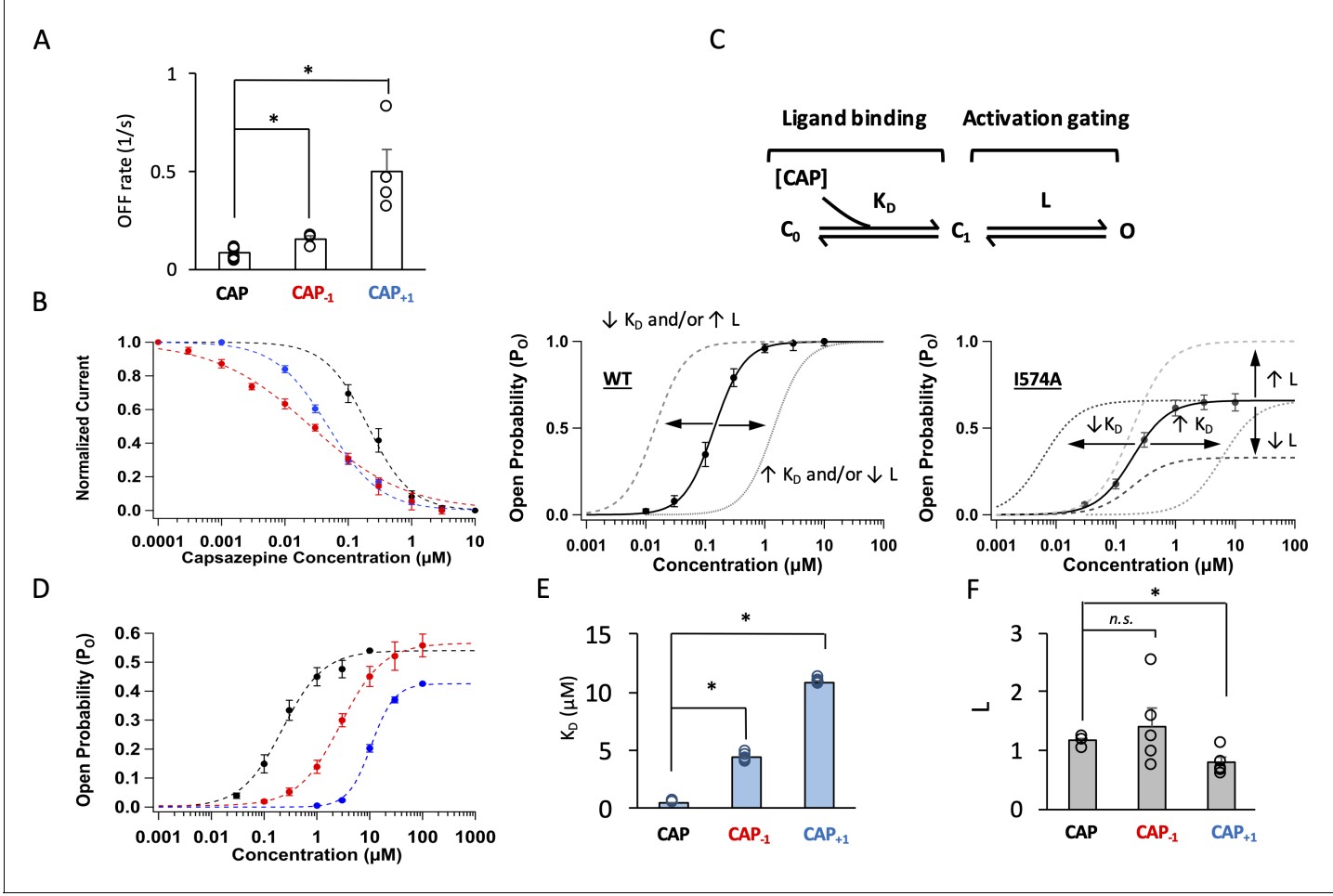

**Figure 4.** Modifying the neck of capsaicin lowers the binding affinity and allosteric constant for gating. (A) Comparison of the OFF rates. (B) Concentration-response curves of capsazepine inhibition, with a Hill function fit superimposed. $n$ = 3–5 cells. (C) Gating scheme of capsaicin ligand binding to TRPV1 and activation gating, with corresponding equilibrium constant $K$ and $L$, respectively. Graphical representation of the challenge for interpretation of changes in $EC_{50}$ of the WT channels when $P_o$ is high (bottom left). The I574A mutation reduces the maximal $P_o$, allowing for the differentiation between changes in $K$ and $L$. (D) Concentration-response curves of TRPV1 I574A in the presence of capsaicin or its analogs. $n$ = 4–6 cells. (E) Comparison of $K$ and (F) $L$ values for capsaicin and its analogs.

The online version of this article includes the following figure supplement(s) for figure 4:

**Figure supplement 1.** Capsaicin analogs exhibited a more rapid off rate.

**Figure supplement 2.** Capsaicin analogs display a reduced binding affinity.

**Figure supplement 3.** Activation of TRPV1 I574A mutant by capsaicin and its analogs.

determined at saturating ligand concentrations as $L = Po\_max/(1 - Po\_max)$. $K_D$ represents the dissociation constant for the ligand-binding step and can be estimated from $EC_{50}$ and $L$ as $K_D = EC_{50}(1 + L)$. For the wildtype (WT) channels, the $Po\_max$ value approached unity in the presence of each capsaicin analog (*Figure 2*), a situation that renders the estimation of $L$ and $K_D$ unreliable as the denominator in the first equation, $(1 - Po\_max)$, becomes very small at saturating ligand concentrations. A manifestation of this limitation is that a shift of the concentration-response curve can be resultant of a change in either $L$ or $K_D$ (*Figure 4C*, bottom left panel; *Zagotta, 2015*). We have previously reported that introducing a background I574A mutation to TRPV1 helped circumvent this problem by reducing $Po\_max$ to about 0.5 (*Figure 4C*, bottom right panel; *Figure 4—figure supplement 3*; *Yang et al., 2015*). With I574A-containing channels, we found that both Cap$_{-1}$ and Cap$_{+1}$ produced a concentration-response curve that was right-shifted, consistent with a higher $K_D$ and weaker binding, a conclusion consistent with unbinding analyses described above (*Figure 4D and E*). In addition, Cap$_{+1}$ exhibited a lower efficacy compared to that of capsaicin, reflecting that $L$

was much reduced for $Cap_{+1}$ (*Figure 4D and F*). In summary, both $Cap_{-1}$ and $Cap_{+1}$ bind less tightly to TRPV1, and $Cap_{+1}$ is also less powerful in inducing the activation allosteric transition upon binding. These changes in gating properties are not simply correlated to the length of the agonists.

## The permissive conformations of the ligand-binding pocket

To better understand how binding of capsaicin and its analogs stabilizes the permissive conformations of the ligand-binding pocket, we examined the binding pocket structure of the top 30 models. Interestingly, while both $CAP_{-1}$ and $CAP_{+1}$ adopted drastically different poses, the binding pocket itself appeared to be in very similar conformations among the top models. The ligand-bound pocket conformations resembled that of the capsaicin-bound cryo-EM structure (PDB_ID 3J5R) (*Liao et al., 2013*). The similarity could be best seen when we aligned each of the top 30 models for $CAP_{-1}$ or $CAP_{+1}$-bound structures to the S4 segment of 3J5R, and calculated RMSD per residue along S4, S4-S5 linker, S5 and S6 (*Figure 5Aa nd B*). The maximal deviation from the capsaicin-bound conformation was within 1 Å. Interestingly, it is noticed that residues with a higher degree of deviation were all clustered around T551, E571, and T671, the three sites of ligand interactions, indicating the occurrence of local side-chain adaptations to accommodate each ligand molecule. The observation that structurally distinct ligands stabilize a single permissive conformation in the TRPV1 ligand-binding pocket suggests that there is a dominant low-energy conformation when a vanilloid molecule replaces the endogenous lipid molecule.

We also examined the channel pore conformation of each of the top 30 ligand-bound models for $Cap_{-1}$ and $Cap_{+1}$. The pore conformations were nearly identical and resembled that seen in the capsaicin-bound cryo-EM structure (*Figure 5—figure supplement 1A and B*, left panels). The pore diameter profiles of these models exhibited a maximal deviation from that of the capsaicin-bound cryo-EM structure by less than 0.4 Å (*Figure 5—figure supplement 1A and B*, right panels). Functional analyses of $Cap_{-1}$ and $Cap_{+1}$ induced currents revealed similar cation selectivity and permeability as capsaicin induced currents (*Figure 5—figure supplement 2*).

The similarities between the ligand-bound conformations at the ligand-binding pocket for capsaicin, $Cap_{-1}$, and $Cap_{+1}$ offer a structural support for TRPV1 ligand activation as an allosteric process (*Latorre et al., 2007*; *Matta and Ahern, 2007*; *Cao et al., 2014*; *Yang et al., 2010*; *Jara-Oseguera and Islas, 2013*). The observed functional differences in $Cap_{-1}$ or $Cap_{+1}$ induced currents are results of combinatory effects of $K_D$ and $L$ as defined in the gating kinetic model shown in *Figure 4C*. This gating model is a simplified form of an allosteric model under the condition that unliganded channel openings are rare, which has been satisfactorily confirmed for the mouse TRPV1 channels (*Yang et al., 2018*; *Yang et al., 2020*). Eyring energy profiles for the three agonists at 10 µM based on functional data are shown in *Figure 5C*. These plots illustrate that capsaicin is a full agonist for TRPV1 as it strongly stabilizes the open state; in comparison, $Cap_{-1}$ and $Cap_{+1}$ are weaker agonists with a less stable open state. Since these agonists can bind in alternative poses, the plots must reflect the combination of these poses and their relative stabilities.

In summary, the TRPV1 ligand-binding pocket allows distinct ligand binding poses but takes a single permissive conformation. Therefore, allosteric activation of TRPV1 can be induced by structurally similar capsaicin analogs in distinct ways (*Figure 5D*), with the strength of inducing a conformational change in the ligand-binding pocket exhibiting correlation with their size. TRPV1 is known to be activated by diverse agonists, many of which remain less studied. In particular, a number of endovanilloids such as anandamide and 12-HEPTE have been reported to target TRPV1 to produce some of their physiological functions that could be antagonized by capsazepine or iodo-resiniferatoxin (*Lam et al., 2007*; *Hwang et al., 2000*; *Jennings et al., 2003*). It remains to be tested whether findings reported here are applicable to other TRPV1 agonists.

## Discussion

As an allosteric protein, TRPV1 activates through coupled conformational changes in the ligand-binding pocket and the channel pore (*Latorre et al., 2007*; *Matta and Ahern, 2007*; *Cao et al., 2014*; *Yang et al., 2010*; *Jara-Oseguera and Islas, 2013*). The stability of an open pore, which determines the $P_o$ level, is linked to the stability of the permissive conformation of the ligand-binding pocket, which is in turn determined by molecular interactions between ligand and its surrounding channel residues. Well-studied ligand-gated ion channels are found to use distinct strategies for

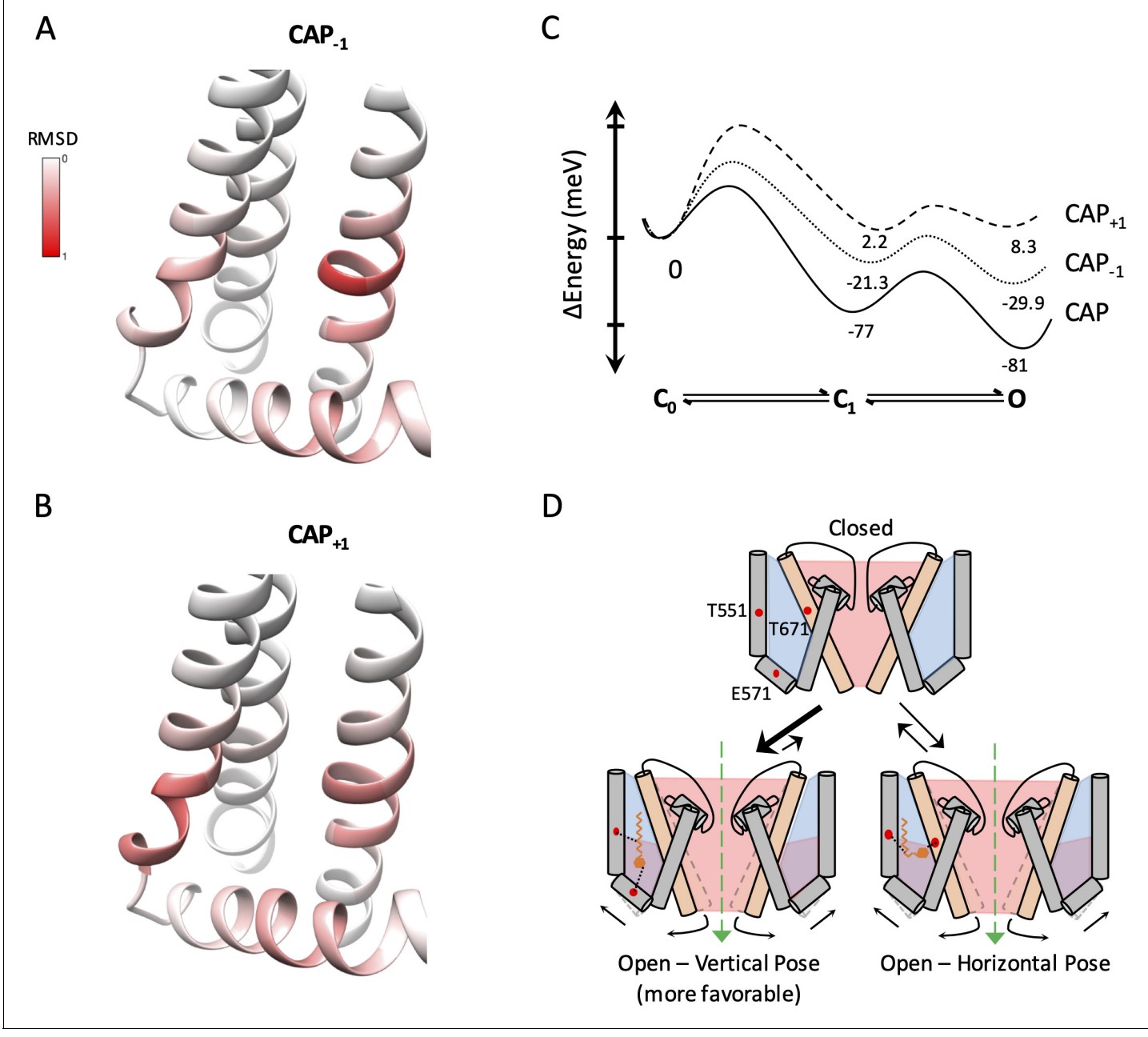

**Figure 5.** Capsaicin analogs elicit a structurally similar permissive state of the TRPV1 ligand-binding pocket. (A and B) Comparisons of the ligand-binding pocket permissive conformations induced by CAP$_{-1}$ (A) or CAP$_{+1}$ (B) to the cryo-EM structure of capsaicin-bound state (3J5R). The backbone RMSD of the top 30 models are presented. (C) Eyring energy profiles of capsaicin and its analogs. The concentration of each ligand was taken as 10 µM. (D) Cartoon summary of TRPV1 activation by capsaicin and its analogs. Blue represents electropositive areas and red represents electronegative areas.

The online version of this article includes the following figure supplement(s) for figure 5:

**Figure supplement 1.** Pore radii of Cap$_{-1}$ (A) and Cap$_{+1}$ (B) models (left) with its distribution plot (right).
**Figure supplement 2.** TRPV1 currents induced by Cap and its analogs exhibit similar ion permeation properties.
**Figure supplement 3.** Electrostatic potential distribution of TRPV1 in the capsaicin-bound state, with top and bottom views.
**Figure supplement 4.** Electrostatic potential of TRPV1 ligand-binding pocket.
**Figure supplement 5.** Electrostatic potential of capsaicin and its analogs.
**Figure supplement 6.** Electrostatic potential maps of capsaicin and halogenated capsaicin analogs.

ligand discrimination (*Hille, 2001*; *Zheng and Trudeau, 2015*). Outstanding examples include glutamate receptors, for which structurally distinct ligands bind in similar poses and induce graded conformational changes in the ligand-binding domain to produce graded channel activities, and these multiple ligand-binding domain conformations define partial and full agonists (*Jin et al., 2003*; *Armstrong and Gouaux, 2000*). Using a set of varying size but chemically similar agonists, we found in TRPV1 the existence of very different ligand-binding poses inside the ligand-binding pocket, consistent with previous studies of natural TRPV1 activators (*Yin et al., 2019*; *Dong et al., 2019*). Nonetheless, these distinct ligand-binding poses all support a single permissive ligand-binding pocket conformation. These observations argue for a single stable permissive conformation for the TRPV1 ligand-binding pocket, one that the cryo-EM structures of capsaicin- and RTX-bound channels demonstrated (*Gao et al., 2016*; *Liao et al., 2013*).

Ligands that better stabilize the ligand-binding pocket in its permissive conformation are expected to yield a more stable pore conformation and induce stronger channel activation. Conformational changes in the TRPV1 pore are currently under intensive investigation and are thought to involve movements of S6 and the selectivity filter (*Cao et al., 2013*; *Gao et al., 2016*; *Yang et al., 2018*; *Jara-Oseguera et al., 2019*). Our present study focused on the conformational change in the ligand-binding pocket, in particular movement of the S4-S5 linker toward S4 (*Figure 1B*). The S4-S5 linker movement is promoted by binding of capsaicin as it bridges S4 and the S4-S5 linker with two hydrogen bonds and extensive VDW interactions (*Yang et al., 2015*; *Elokely et al., 2016*). Our results from structurally related ligands revealed how the molecular structure of capsaicin makes it a very potent agonist for TRPV1: not only does it have a distribution of hydrophobic and hydrophilic groups that matches complementarily to the electrostatic distribution of the ligand-binding pocket, but also the separation between the hydrogen bond-forming groups (an amide in the neck and a hydroxyl group in the head) matches perfectly the distance between T551 in S4 and E571 in the S4-S5 linker in the permissive conformation. Contributions from both the size and electrostatic properties are highlighted by the two capsaicin analogs tested in the present study.

## The size factor that contributes to agonist binding

When bound to TRPV1, capsaicin induces a movement of the S4-S5 linker toward S4 by 1.3 Å. Capsaicin stabilizes the activated conformation of the S4-S5 linker partially by the two hydrogen bonds between capsaicin and TRPV1. $Cap_{-1}$ and $Cap_{+1}$, being about 1.5 Å shorter or longer than capsaicin, respectively, can form these hydrogen bonds. However, the size mismatch likely causes these agonists to be not as stable as capsaicin in this binding pose. Therefore, they explore other low-energy binding poses. The alternative binding poses are not as stable as the vertical pose of capsaicin, as the computational modeling results revealed (*Figure 3*). This explains why $Cap_{-1}$ and $Cap_{+1}$ are less potent agonists compared to capsaicin. The existence of alternative binding poses for capsaicin analogs is not surprising; indeed, when the two hydrogen bond-forming residues, T551 and E571, were simultaneously mutated to a hydrophobic residue to prevent the vertical pose, capsaicin became a less potent agonist but nonetheless could still induce strong channel activation at high concentrations (*Yang et al., 2015*). This observation suggested that capsaicin must also be able to bind in an alternative pose(s) when the most favorable binding pose is prohibited.

The differential effects of changing agonist length on $K_D$ and $L$ as observed in $Cap_{-1}$ and $Cap_{+1}$ do not directly correlate with differences in molecular size, suggesting that additional factors must be involved. While it remains to be explored what other factors are present, we speculate that the electrostatic potential distribution inside the ligand-binding pocket might play an important role.

## Potential electrostatic influence on agonist binding

Our study revealed that the binding poses of the two capsaicin analogs are stabilized by both polar and nonpolar interactions. We identified multiple polar residues in the ligand-binding pocket that mediate agonist-channel interactions in these poses. Mutations to these polar residues substantially reduced ligand activation. Nonpolar residues also contribute to agonist-channel interactions (*Figure 3B and E*). While VDW interactions are in general harder to pinpoint and prove in functional tests, we have previously shown that progressively shortening capsaicin's hydrophobic tail led to incremental right-shifts of the concentration-response curve, with an associated increase in $EC_{50}$ by as much as three orders of magnitude (*Yang et al., 2015*). In the present study, we observed from

Cap$_{+1}$ that the horizontal pose prevented the tail from going deep into the highly hydrophobic upper region of the ligand-binding pocket. The resultant loss of VDW interactions likely contributes to the reduced potency of this analog. Indeed, F544 in the hydrophobic upper region is found to contribute substantially to VDW interaction with the tail of capsaicin (*Yang et al., 2015*) and Cap$_{-1}$ (*Figure 3B*); a hydrophilic residue at the equivalent position of TRPV2 (S498) needed to be mutated to a phenylalanine in order to introduce capsaicin binding to the capsaicin-insensitive TRPV2 (*Yang et al., 2016*).

The electrostatic properties of the ligand-binding pocket likely contribute substantially to capsaicin binding. The TRPV1 cryo-EM structures reveal that the ion permeation pathway is electro-negative, allowing the passage of cations (*Figure 5—figure supplement 3*), like other cation channels (*Li et al., 2011*; *Noskov et al., 2004*; *Nimigean et al., 2003*). The ligand-binding pocket, on the other hand, is mostly electro-neutral to accommodate a native lipid molecule in the closed conformation (*Gao et al., 2016*; *Figure 5—figure supplement 4, left*) and capsaicin in the open conformation (*Cao et al., 2013*; *Liao et al., 2013*; *Figure 5—figure supplement 4, right*). While the overall polarized electrostatic potential distribution inside the ligand-binding pocket remains undisturbed, local changes clearly occur as the ligand-binding pocket moves from apo to permissive state. Because the electrostatic potential distribution of capsaicin or its analog are distinct (*Figure 5—figure supplement 5*), the electrostatic influence on them is likely to be different. Of particular notice is the presence of a side portal in the ligand-binding pocket that is directly accessible to the electro-negative pore (*Figure 5—figure supplement 3*, right panel). As the presence of this portal, not observed in the apo state, is associated with the conformational changes during activation, ligands interacting distinctly with it are expected to exert an energy bias toward allosteric activation. Indeed, introducing a single halogen atom (Cl, Br or I) to the head of either capsaicin or RTX could turn these TRPV1 agonists into competitive antagonists (*Appendino et al., 2003*; *Wahl et al., 2001*; *Appendino et al., 2005*; *Figure 5—figure supplement 6*). Comparative studies utilizing structurally related compounds are expected to further reveal the structural mechanisms underlying potent capsaicin activation of TRPV1.

## Existence of a single permissive ligand-binding pocket conformation

Structural studies suggest that the TRPV1 ligand-binding pocket moves into a stable permissive conformation when capsaicin or RTX is bound (*Gao et al., 2016*; *Liao et al., 2013*). These ligands are thought to interact with both S4 and S4-S5 linker (*Gao et al., 2016*; *Yang et al., 2015*). Cap$_{-1}$ and Cap$_{+1}$ in the present study and plant-derived natural ligands in previous studies (*Yin et al., 2019*; *Dong et al., 2019*) can bind in another pose without a direct interaction with S4-S5 linker, yet they produce a very similar permissive conformation of the ligand-binding pocket. Therefore, in TRPV1, the ligand recognition mechanism has evolved following the allosteric principle. A full agonist and a partial agonist are distinguished by their ability to stabilize the permissive conformation, governed by distinct ligand-receptor interactions inside the ligand-binding pocket (*Figure 5D*). These structural insights offer a framework for understanding ligand-induced conformational coupling during activation gating, and further support the notion that TRPV1 functions as an allosteric protein.

## Materials and methods

**Key resources table**

| Reagent type (species) or resource | Designation | Source or reference | Identifiers | Additional information |
|---|---|---|---|---|
| Cell line (*Homo sapiens*) | HEK 293 | ATCC | Cat #: CRL-1573 | |
| Chemical compound, drug | Capsaicin | Sigma | Cat #: M2028 | |
| Chemical compound | (*E*)—8-methylnon-6-enoyl chloride | TCI America | Cat #: 95636-02-5 Product: M1826 | |

*Continued on next page*

*Continued*

| Reagent type (species) or resource | Designation | Source or reference | Identifiers | Additional information |
|---|---|---|---|---|
| Chemical compound | 4-(2-aminoethyl)−2-methoxyphenol | TCI America | Cat #: 7149-10-2 Product: A2330 | |
| Chemical compound | 4-amino-2-methoxyphenol | TCI America | Cat #: 52200-90-5 Product: A2883 | |
| Software, algorithm | IgorPro | IgorPro (https://www.wavemetrics.com/) | Version 8 | |
| Software, algorithm | Rosetta | Rosetta (https://www.rosettacommons.org/) | Version 3.10 | |
| Software, algorithm | Chimera | UCSF Chimera (https://www.cgl.ucsf.edu/chimera/) | Version 1.14 | |
| Software, algorithm | VMD | VMD (https://www.ks.uiuc.edu/Research/vmd/) | Version 1.9.3 | |
| Software, algorithm | GAMESS | GAMESS (https://www.msg.chem.iastate.edu/GAMESS/) | Version Sept. 30,2018 R3 | |
| Software, algorithm | MultiWFN | MultiWFN (http://sobereva.com/multiwfn/) | Version 3.6 | |
| Software, algorithm | Avogadro | Avogadro (https://avogadro.cc/) | Version 1.2.0 | |
| Software, algorithm | OpenEye OMEGA | OpenEye OMEGA (https://www.eyesopen.com/omega) | Version 2.4.3 | |
| Software, algorithm | QuB | QuB (https://www.qub.buffalo.edu/) | Version 2.0.0.30 | |

## Molecular biology and cell transfection

Murine TRPV1 cDNA was used for this study. A copy of the enhanced yellow fluorescent protein (eYFP) cDNA was fused to its C-terminus to facilitate identification of transfected cells, as previously described (*Cheng et al., 2007*). This construct was cloned into the pEYFP-N3 expression vector. Point mutations were performed by QuickChange II mutagenesis kit (Agilent Technologies) and confirmed by sequencing.

HEK293T cells were purchased from American Type Culture Collection (ATCC) and cultured in Dulbecco's modified eagle medium (DMEM) supplemented with 20 mM L-glutamine and 10% fetal bovine serum. No mycoplasma contamination was detected using Lonza MycoAlert Mycoplasma Detection Kit and the cell line was authenticated by ATCC using STR profiling. Cells were transiently transfected with cDNA using Lipofectamine 2000 (Life Technologies), and experiments were performed 18–24 hr after transfection.

## Chemicals

(*E*)-*N*-(4-Hydroxy-3-methoxyphenethyl)-8-methylnon-6-enamide (Cap$_{+1}$) and (*E*)-*N*-(4-hydroxy-3-methoxyphenyl)-8-methylnon-6-enamide (Cap$_{-1}$) were synthesized by reacting (*E*)-8-methylnon-6-enoyl chloride with 4-(2-aminoethyl)-2-methoxyphenol or 4-amino-2-methoxyphenol, respectively (Supplementary Notes). The identity and purity of both Cap$_{+1}$ and Cap$_{-1}$ were confirmed by $^1$H, $^{13}$C

NMR and high-resolution mass spectroscopy (HRMS; Supplementary Notes). Chemical structures were drawn using ChemDraw 19.0 (Perkin Elmer). All the other chemicals were purchased from Sigma-Aldrich.

## Electrophysiology

Patch-clamp recordings were performed using a HEKA EPC10 amplifier with PatchMaster software (HEKA). Electrical signals were filtered at 2.9 kHz and sampled at 10 kHz. For macroscopic recordings, patch pipettes were fashioned from borosilicate glass and fire polished within the range of 3–5 MΩ. Both bath and pipette solutions contained 140 mM NaCl, 0.2 mM EGTA, and 5 mM HEPES (pH 7.2) unless otherwise stated. All recordings were performed at room temperature (~22℃) in either whole-cell or inside-out configurations. Currents were recorded with a protocol consisting of a holding potential at 0 mV, a step to +80 mV for 200 ms followed by a step to −80 mV for 200 ms before returning to the holding potential. Solutions were delivered in separate tubes onto the cell or excised-patch membrane using a rapid solution changer (RSC-200, Bio-Logic) to prevent solution mixing. Single-channel recordings under the inside-out configuration were performed 6–8 hr after transfection to maximize the chance of obtaining patches containing a single channel. Patch pipettes were fire-polished to resistance of 8–12 MΩ. All single-channel recordings were obtained with the membrane potential clamped to +80 mV. Conductance (g) was calculated at +80 mV. Single channel recordings were analyzed with Igor Pro 8 (WaveMetrics). For reversal potential ($E_{rev}$) measurements, 140 mM NaCl was replaced with 70 mM $MgCl_2$ or 70 mM $CaCl_2$ in the intracellular solution, with EGTA excluded. Currents from inside-out patches were recorded with a 500 ms voltage ramp protocol consisting of a 0 mV holding potential, a pre-step to −100 mV for 100 ms, a voltage ramp from −100 mV to +100 mV for 500 ms, and a post-step at +100 mV for 100 ms before returning to the holding potential. Permeability ratios were determined using the $E_{rev}$ measurements with the Goldman-Hodgkin-Katz equation.

## Data analysis

Macroscopic and single channel recordings were analyzed using Igor Pro 8 (Wavemetrics). Current amplitude was fitted to a Hill equation to generate concentration-response curves which were used to obtain estimates of the $EC_{50}$, $IC_{50}$, and the Hill coefficient. The OFF rate was determined by fitting the time course of tail current from inside-out recordings at +80 mV to a single exponential function.

Single-channel recordings were analyzed using all-point histograms generated at +80 mV where a double-Gaussian function was used to fit the histograms. Event detection was performed using the threshold-crossing method to produce idealized event lists. From the peak values, single-channel current amplitude was determined for each compound. Single channel open probability was determined using QuB software. A dead time of 0.3 ms was imposed and traces were idealized using the 50% amplitude threshold-crossing method. Patches with more than one channel were discarded. Open probability of macroscopic recordings of each analog was determined by normalizing to that of saturating concentrations of capsaicin (10 μM). $CAP_{-1}$ macroscopic open probability was determined by first correcting for the current difference due to single-channel amplitude.

All statistical values are given as mean ± SEM. Statistical significance was determined using the Student's t test. *, $p < 0.05$.

## Molecular docking

Docking of capsaicin analogs was performed using RosettaLigand and RosettaScripts, an application within the Rosetta modeling software (version 3.8). The TRPV1 capsaicin-bound cryo-EM structure (PDB: 3J5R) was relaxed in a membrane environment using RosettaMembrane. Because the current version of Rosetta cannot handle explicit lipids, annular lipids or PIP2 were not modeled. Water molecules were also not modeled; their involvements in capsaicin binding were previously studied by *Elokely et al., 2016*. The analogs were initially placed within the vanilloid binding pocket and constrained within a 5 Å diameter sphere where it could freely move during the model improvement process. For each analog, 200 conformers were generated using Open Eye OMEGA. As an initial phase of the docking process, both the channel and the ligand were represented coarsely in the centroid-mode before a Monte Carlo search was prompted. Once complete, the lowest energy structure was selected for further high-resolution refinement. Here, the centroid atoms were replaced

with their respective side-chain atoms where its positions were randomly perturbed within a Gaussian distribution. The conformations were energy-minimized and the rotamers were optimized for each position with RotamerTrials. Side-chain optimization was carried out using the full side-chain packing algorithm followed by a Metropolis criterion check. The backbone of TRPV1 was restricted to a maximum of 5 Å of displacement from the native structure. Initially, 30,000 models were generated which were then screened for the lowest energies. The top 1% lowest energy models were further screened for binding energy between the ligand and the channel. From these, the top 30 models were identified as candidates for further detailed analyses. To determine potential atomic interactions between the ligand and the channel, the binding energies were decomposed into two main categories – hydrogen bonding and Van der Waals (VDW) energies on a per residue basis using Rosetta's residue_energy_breakdown function. All molecular graphics of TRPV1 and its agonists were rendered using UCSF Chimera 1.14 unless otherwise stated.

RMSD measurements were calculated using RosettaScripts. Briefly, the top 1% lowest energy models were analyzed using the SimpleMetricsFilter within RosettaScripts. All measurements were calculated using the capsaicin-bound model as the reference. RMSD was calculated per residue and plotted using UCSF Chimera. Pore radius was calculated using the HOLE program version 2.0 (*Smart et al., 1996*). The top 1% lowest energy models were analyzed and then visualized in VMD 1.9.3.

## Electrostatic potential analysis

Avogadro was used to generate and optimize the geometry of the compounds and create input files. Using these input files, GAMESS software (*Li Manni et al., 2014*; *Gagliardi et al., 2017*; *Carlson et al., 2015*) was used to calculate the wavefunction for each compound. The output files were then input into MultiWFN (*Lu and Chen, 2012*) to calculate the electrostatic potential of individual atoms on the ligands and its localization. Electrostatic potential maps of each compound were visualized in VMD. Electrostatic potential of TRPV1 was calculated by Adaptive Poisson-Boltzmann Solver (APBS) in UCSF Chimera.

## Acknowledgements

We are grateful for the assistance and insights from current and former members of the Zheng lab. Work in the Zheng and Yarov-Yarovoy labs was partially supported by funding from NIH (R01NS103954 and R01GM132110). SV was partially supported by a fellowship from AHA (16PRE29340002). VS and HW were supported by the CounterACT Program, National Institutes of Health Office of the Director, and the National Institute of Neurological Disorders and Stroke (U54NS079202).

## Additional information

### Funding

| Funder | Grant reference number | Author |
| --- | --- | --- |
| National Institutes of Health | R01NS103954 | Vladimir Yarov-Yarovoy Jie Zheng |
| American Heart Association | 16PRE29340002 | Simon Vu |
| National Institutes of Health | R01GM132110 | Vladimir Yarov-Yarovoy Jie Zheng |
| CounterACT Program (National Institute of Health Directors Office and National Institute of Neurological Disease and Stroke) | U54NS079202 | Vikrant Singh Heike Wulff |

The funders had no role in study design, data collection and interpretation, or the decision to submit the work for publication.

## Author contributions
Simon Vu, Data curation, Formal analysis, Funding acquisition, Investigation, Methodology, Writing - review and editing; Vikrant Singh, Data curation, Formal analysis, Funding acquisition, Methodology; Heike Wulff, Resources, Supervision, Funding acquisition, Methodology, Project administration, Writing - review and editing; Vladimir Yarov-Yarovoy, Resources, Software, Supervision, Funding acquisition, Methodology, Project administration, Writing - review and editing; Jie Zheng, Conceptualization, Resources, Software, Supervision, Funding acquisition, Validation, Investigation, Visualization, Methodology, Writing - original draft, Project administration, Writing - review and editing

## Author ORCIDs
Simon Vu (iD) https://orcid.org/0000-0002-1529-8220
Vladimir Yarov-Yarovoy (iD) http://orcid.org/0000-0002-2325-4834
Jie Zheng (iD) https://orcid.org/0000-0002-4161-627X

## Decision letter and Author response
Decision letter https://doi.org/10.7554/eLife.62039.sa1
Author response https://doi.org/10.7554/eLife.62039.sa2

# Additional files

## Supplementary files
• Supplementary file 1. Description of synthesis, $^1$H NMR, and $^{13}$C NMR of capsaicin analogs. Rosetta commands and scripts for docking.

• Transparent reporting form

## Data availability
All data generated or analyzed during this study are included in the manuscript and supporting files.

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
