## [Decision Letter]

**Acceptance summary:**

TRPV1 channels are sensitive to a wide variety of endogenous and exogenous chemical and physical stimuli. This paper employs subtle analogs of the toxin agonist capsaicin, the active component in chilli peppers, to show that despite vastly different ligand binding poses, the capsaicin analogs elicit very similar conformations of the TRPV1 ligand-binding pocket. The work thereby provides important insight on how ligands modulate TRPV1 activity.

**Decision letter after peer review:**

Thank you for submitting your article "Novel Capsaicin Analogs as Molecular Rulers to Define the Permissive Conformation of TRPV1 Ligand-Binding Domain" for consideration by *eLife*. Your article has been reviewed by three peer reviewers, and the evaluation has been overseen by a Reviewing Editor and Richard Aldrich as the Senior Editor. The following individuals involved in review of your submission have agreed to reveal their identity: Vincenzo Carnevale (Reviewer #2); Vera Y Moiseenkova-Bell (Reviewer #3).

The reviewers have discussed the reviews with one another and the Reviewing Editor has drafted this decision to help you prepare a revised submission.

Summary:

The manuscript by Vu et al. reports on an interesting investigation of the structural underpinnings of TRPV1 activation by capsaicin analogs. In particular, thanks to a combination of chemical synthesis, kinetic modeling based on electrophysiology and molecular modeling, the authors conclude that two possible binding modes stabilize the gating-permissive conformation of the ligand-binding domain. This topic is relevant given that the microscopic mechanism of activation of TRPV1 is still not fully understood and the fact that novel TRPV1 channel modulators are sought-after for their potential use as pain killers. The experiment and computational approaches are sound and are generally carried out with the appropriate scientific rigor and the manuscript is well written. However, additional data is needed to confirm the validity of the structural model used. Further, the authors need to carefully revise the text with regard to the general applicability of their findings and expand on the discussion of a number of points, see below.

Essential revisions:

1) The structural model employed for the calculations (PDB: 3J5R) is a key concern. This very early structure of TRPV1 is not of high resolution/quality. There are several more recent studies and the functional state corresponding to this particular structure is still somewhat controversial. Therefore, using this model as a bona-fide open state raises important concerns. There is an abundance of recent structural information for members of the TRPV subfamily, including TRPV1 (some of these channels have been solved in both the open and closed conformations and thus provide suitable templates). It is therefore important to validate the robustness of the present findings by using a different model of the open state.

2) Capsaicin is a powerful tool to study the activation mechanism of TRPV1. However, this molecule represents toxin activation of the channel, and the comparison with classical neurotransmitters is therefore not relevant (Introduction). The assumption that all TRPV1 agonists acting through the capsaicin binding site result in the same permissive open confirmation should be tested. Ideally, this should be conducted with endogenous activators that do not have a similar chemical structure as capsaicin, such as anandamide or 12-HEPTE. If such data cannot be generated in a reasonable time frame, the authors need to carefully emphasize throughout the manuscript that their present findings are only relevant to capsaicin and capsaicin analogs (including the title).

3) Previous work on TRPV1 based on all atom molecular dynamics have shown that the interaction between capsaicin and the vanilloid binding site are partially mediated by tightly bound water molecules (see Elokely et al., 2016). Have the authors considered this possibility for the binding of capsaicin and the two analogs? It is our understanding that the Rosetta modeling strategy employed here may not allow for explicit description of such water molecules. If this is the case, the authors should acknowledge this limitation and, consequently, provide sufficient context with regards to the computational results in the Discussion.

4) Another important concern relates to the fact that annular lipids seem to play a functional role in TRP channels in general and in the TRPV family in particular. Several structural determinations of members of this evolutionary family have found that lipids bind to crevices adjacent or overlapping with the vanilloid binding site in a state-dependent fashion (see Singh et al., Nat. Str. and Mol. Biol. 2019). Assuming that a similar mechanism is at work in TRPV1, are these lipid molecules found in the open state of the channel compatible with both the binding sites? Again, for the purpose of providing a fair a complete context to the results, including limitations, pitfalls and possible alternative explanations, it is important that the authors extend the structural analysis to take into consideration this aspect as well.

5) In part related to the previous point is the structural role of PIP2, which is a well-established positive regulator of TRPV1. Current experimental information indicates that this lipid molecule binds to the S4-S5 linker in a conformation that seemingly interferes with one of the two binding modes proposed in this manuscript (the so-called horizontal pose). Is that the case? If so, what are the functional consequences of this?

6) The term ligand-binding domain is somewhat misleading, given that the agonists do not bind to a discrete protein domain, but rather to part of the transmembrane core. There is no gain in introducing a new nomenclature and the authors should therefore use standard terminology (vanilloid binding site, ligand-binding pocket or similar).

[Editors' note: further revisions were suggested prior to acceptance, as described below.]

Thank you for resubmitting your work entitled “Novel Capsaicin Analogs as Molecular Rulers to Define the Permissive Conformation of mouse TRPV1 Ligand-Binding Pocket" for further consideration by *eLife*. Your revised article has been evaluated by Richard Aldrich (Senior Editor) and a Reviewing Editor.

The manuscript has been improved but there are some remaining issues that need to be addressed before acceptance, as outlined below:

In light of the observed differences between 3J5R and 5IRX, as well as the missing water, annular lipids and PIP2 (currently mentioned in Materials and methods), the authors are asked to mention these aspects more explicitly in the main text. Either as part of the results or as a brief separate section in the discussion. This will not take away from the impact of their study but will help the reader to put the work into context.

---

## [Author Response]

Essential revisions:1) The structural model employed for the calculations (PDB: 3J5R) is a key concern. This very early structure of TRPV1 is not of high resolution/quality. There are several more recent studies and the functional state corresponding to this particular structure is still somewhat controversial. Therefore, using this model as a bona-fide open state raises important concerns. There is an abundance of recent structural information for members of the TRPV subfamily, including TRPV1 (some of these channels have been solved in both the open and closed conformations and thus provide suitable templates). It is therefore important to validate the robustness of the present findings by using a different model of the open state.

The reviewer’s point is well-taken. We have in recent years studied binding of capsaicin (Yang et al., 2015), ginger compounds (Yin et al., 2019), and piperine (Dong et al., 2019) to TRPV1. These studies, initiated shortly after the report of 3J5R, were all based on this structural model. To be consistent, especially for direct comparison between binding of capsaicin and that of its analogs, we decided to use 3J5R in the present study as well. For modeling, we first relaxed 3J5R structure in a membrane environment using the RosettaMembrane application, as described in our report on capsaicin binding (Yang et al., 2015). (This information is provided in the Materials and methods section of the revised manuscript.)

The 3J5R structure represents a ligand-bound, closed channel (perhaps in a pre-open state?). The newer TRPV1 structure in complex with both RTX and DkTx, 5IRX, represents a ligandbound, open channel. When compared directly, their ligand-binding pocket conformations are similar whereas the pore region conformations, especially those of S5 and pore helix, are rather different. The comparison, as well as testing results, see Author response image 1, argues for the applicability of 3J5R for ligand binding study as in the present project. In response to the reviewer’s comment, we have revised the manuscript to tone down the part on pore conformation of CAP analog-bound channels and moved Figure 5C and D into a supplementary figure.

**Author response image 1. sa2fig1:** Comparisons between 3J5R and 5IRX. (Left) Per-residue structural deviation is plotted for the RMSD in Å between these two structures in the ligand-binding pocket. (Right) Structural overlay of 3J5R (cyan) and 5IRX (brown) between S1 and TRP helix.

We further ran a quick check on binding of capsaicin to 5IRX, using the same routines in Rosetta as we have applied to 3J5R in the present study. Initial results from 5IRX closely resembled what we have observed from 3J5R, with an unambiguous head-down tail-up pose secured by clearly identifiable neck-to-T551 interactions. Encouraged by the reviewer’s suggestion, we are initiating a thorough investigation of binding by capsaicin analogs to 5IRX and will verify modeling results with functional tests. While this effort will take much longer to complete, we will find ways to share the results once the study is completed.

2) Capsaicin is a powerful tool to study the activation mechanism of TRPV1. However, this molecule represents toxin activation of the channel, and the comparison with classical neurotransmitters is therefore not relevant (Introduction). The assumption that all TRPV1 agonists acting through the capsaicin binding site result in the same permissive open confirmation should be tested. Ideally, this should be conducted with endogenous activators that do not have a similar chemical structure as capsaicin, such as anandamide or 12-HEPTE. If such data cannot be generated in a reasonable time frame, the authors need to carefully emphasize throughout the manuscript that their present findings are only relevant to capsaicin and capsaicin analogs (including the title).

Indeed, a number of “endovanilloids” with an overall similar structure to capsaicin have been reported to have high EC_50_/efficacy for TRPV1. Since capsazepine and iodo-resiniferatoxin (both known to bind to the same ligand-binding site, as seen in cryo-EM structures) exhibited competitive inhibitory effects on channel activation by these compounds (e.g., Lam et al., 2007; Hwang et al., 2000), they are assumed to bind to the same ligand-binding pocket. These functional findings are consistent with those for piperine and capsazepine that have drastically different molecular structures from capsaicin but bind to the same binding pocket. Given that the structures of endovanilloids deviate further from capsaicin than the analogs used in the present study, it is anticipated that they might bind differently, though modeling of these compounds (and functionally confirming models) would require additional efforts beyond the scope of this project. We have modified the manuscript to clarify that the present manuscript focuses on the main conclusion of different ligand binding poses stabilizing the same open confirmation, and that endovanilloids are yet to be tested.

3) Previous work on TRPV1 based on all atom molecular dynamics have shown that the interaction between capsaicin and the vanilloid binding site are partially mediated by tightly bound water molecules (see Elokely et al., 2016). Have the authors considered this possibility for the binding of capsaicin and the two analogs? It is our understanding that the Rosetta modeling strategy employed here may not allow for explicit description of such water molecules. If this is the case, the authors should acknowledge this limitation and, consequently, provide sufficient context with regards to the computational results in the discussion.

As the reviewer correctly pointed out, water molecules must be important for ligand-channel interactions in TRPV1, which was previously studied by Elokely at al., 2016. At the time when we carried out this project, Rosetta did not allow for explicit description of water molecules. While this new function is being added to Rosetta, its performance has not been thoroughly tested in the field or Yarov-Yarovoy/Zheng labs. We have revised our manuscript to state this important limitation of our modeling.

4) Another important concern relates to the fact that annular lipids seem to play a functional role in TRP channels in general and in the TRPV family in particular. Several structural determinations of members of this evolutionary family have found that lipids bind to crevices adjacent or overlapping with the vanilloid binding site in a state-dependent fashion (see Singh et al., Nat. Str. and Mol. Biol. 2019). Assuming that a similar mechanism is at work in TRPV1, are these lipid molecules found in the open state of the channel compatible with both the binding sites? Again, for the purpose of providing a fair a complete context to the results, including limitations, pitfalls and possible alternative explanations, it is important that the authors extend the structural analysis to take into consideration this aspect as well.

We agree that annular lipids are important for TRPV1 functions and wish to better understand their role in ligand activation. However, currently Rosetta cannot handle explicit lipids. We wish to test potential involvements of annular lipids in the future when such functionality is added to Rosetta, and effective functional tests are developed for the same purpose. This limitation is clearly stated in the revised manuscript.

5) In part related to the previous point is the structural role of PIP2, which is a well-established positive regulator of TRPV1. Current experimental information indicates that this lipid molecule binds to the S4-S5 linker in a conformation that seemingly interferes with one of the two binding modes proposed in this manuscript (the so-called horizontal pose). Is that the case? If so, what are the functional consequences of this?

We agree that lipids including PIP2 are important regulators of TRPV1 and may play a role in ligand activation. As mentioned above, modeling of explicit lipids is not currently possible in Rosetta. In the revised manuscript we made clear that PIP2 was not modelled.

6) The term ligand-binding domain is somewhat misleading, given that the agonists do not bind to a discrete protein domain, but rather to part of the transmembrane core. There is no gain in introducing a new nomenclature and the authors should therefore use standard terminology (vanilloid binding site, ligand-binding pocket or similar).

In our original manuscript, we used “ligand-binding domain” and “ligand-binding pocket” interchangeably, in the same sense as LBD has been used for glutamate receptors (whose LBD is formed by two separate segments of GluR, and acetylcholine receptors whose LBD is formed by adjacent subunits). Nonetheless, the reviewer’s point is well-taken. We have now gone through the manuscript to make sure the commonly used term for TRPV1 “ligand-binding pocket” is consistently applied.

[Editors' note: further revisions were suggested prior to acceptance, as described below.]

The manuscript has been improved but there are some remaining issues that need to be addressed before acceptance, as outlined below:In light of the observed differences between 3J5R and 5IRX, as well as the missing water, annular lipids and PIP2 (currently mentioned in Materials and methods), the authors are asked to mention these aspects more explicitly in the main text. Either as part of the results or as a brief separate section in the discussion. This will not take away from the impact of their study but will help the reader to put the work into context.

As suggested, we have added a paragraph in the Results section on limitations of our structural modelling approach.